# Comparison of Hemodynamic Factors Predicting Prognosis in Heart Failure: A Systematic Review

**DOI:** 10.3390/jcm8101757

**Published:** 2019-10-22

**Authors:** Margot Aalders, Wouter Kok

**Affiliations:** Department of Clinical and Experimental Cardiology, Amsterdam Cardiovascular Sciences, Meibergdreef 9, 1105AZ Amsterdam, The Netherlands; m.b.aalders@amsterdamumc.nl

**Keywords:** heart failure, prognosis, cardiac output, preload, afterload

## Abstract

Objectives: We systematically reviewed the literature to address the question of which of the three hemodynamic factors predicts prognosis best in heart failure patients when directly compared to each other: cardiac output, preload or afterload. Methods: Prognostic studies in heart failure (HF) were searched that included at least two of the three hemodynamic variables: (1) cardiac output or cardiac index (CI), (2) preload represented by pulmonary capillary wedge pressure (PCWP) and (3) afterload simplified to systolic blood pressure (SBP). Critical appraisal was done according to the QUIPS format for prognostic studies. The main endpoint was all-cause mortality, which could be combined with other endpoints. We report the number of studies in which CI, PCWP and SBP remained significant prognostic predictors in multivariate analysis. We also assessed whether hemodynamic predictors of prognosis varied in four different HF-populations. Results: Included were 18 studies containing a multivariate analysis. PCWP was an independent predictor of prognosis in 10 of 18 studies, SBP in 3 of 14 studies and CI in none of 18 studies. Results were not specific for any of the HF-populations. Conclusions: A higher PCWP and lower SBP are independent predictors of poor prognosis in HF. In spite of the frequently used concept behind HF, this review demonstrates that CI is not an independent predictor of prognosis in HF.

## 1. Key Questions

### 1.1. What Is Already Known about This Subject?

Prognosis of heart failure patients is known to depend on preload parameters such as capillary wedge pressure, and it is also known that a low blood pressure is predictive of poor prognosis. The prognostic role of cardiac output in heart failure is often assumed, although with little evidence.

### 1.2. What Does This Study Add?

This is the first systematic review on hemodynamic prognostic predictors in heart failure patients. It shows that cardiac output, albeit a univariate predictor of the outcome, is not independently related to outcome in heart failure patients. A higher pulmonary capillary wedge pressure is the most frequently observed independent predictor of poor prognosis.

### 1.3. How Might This Impact on Clinical Practice?

Clinical practice may benefit by recognizing that the treatment goal for prognosis of heart failure primarily consists of decreasing preload.

## 2. Introduction

In the concept of heart failure (HF), decreased cardiac output (CO) is the central problem, followed by and, in part corrected by, an increased preload with secondary increases in afterload due to sympathetic and neurohormonal regulation [1,2,3], where the appropriate treatment goals, therefore, are maintaining an adequate CO and decreasing preload and afterload. Although it may be expected that prognostic models of patients with HF would reflect this concept of HF, CO and invasively measured preload parameters are absent in reviews of prognostic models of HF [4,5]. Better known prognostic factors in patients with HF are age, functional class, systolic blood pressure, renal function, anemia, and plasma levels of sodium, urea, and natriuretic peptides [4,5].

In this study, we aim—for the first time as far as we know—to systematically review the evidence in supporting or denying which of the three factors of cardiac output, preload and afterload is the most important factor in predicting the prognosis of a HF-patient, in various HF-populations.

## 3. Methods

### 3.1. Definitions

We focus on three hemodynamic parameters for left-sided HF, which is the most common form of HF [6]. These are cardiac output (CO), in liters/minute or Cardiac Index (CI) given in liters/minute/m^2^, preload as the pulmonary capillary wedge pressure (PCWP) given in millimeter of mercury (mmHg), and afterload. Afterload is defined as systolic wall-stress of the left ventricle, the product of systolic left ventricular pressure and ventricular diameter, divided by wall thickness. However, in this review, the systolic aortic blood pressure (SBP) was used as a measurable value of afterload in mmHg [7].

Four different HF-populations are distinguished in this review. Acutely decompensated heart failure (ADHF) as defined in guidelines [1], advanced HF being screened for or awaiting heart transplantation (HTX), chronic HF (CHF) patients who attend the outpatient clinic, and HF-patients with preserved ejection fraction (HFPEF). Although HFPEF may be defined using several definitions [1,3], we accepted the definition used by the studies.

### 3.2. Search Strategy

A literature search was performed in the online databases PubMed and Embase, using the terms “Pulmonary Wedge Pressure”, “Blood Pressure”, “Cardiac Output”, “Prognosis” and “Heart Failure” with data until 25 January 2018. The complete search strategy is shown in Appendix A. Included were (1) prognostic studies of adult HF-patients, while studies including children and patient populations after HTX or with implanted LVADs (left ventricular assist devices) were excluded; (2) populations with ≥90 patients; (3) for title and abstract screening, the presence of at least one of three hemodynamic variables was required. For full-text screening, at least two hemodynamic variables had to be present in the analysis of prognosis—allowed in any combination—because of the comparative nature of the review; (4) only studies with invasive hemodynamic measurements of CI and PCWP; SBP could be measured non-invasively. (5) The outcome had to be described as long-term prognosis, which we defined as at least 60 days. The endpoints of prognosis had to be: (all-cause or cardiovascular) mortality, or a combination of mortality with HTX, cardiovascular hospitalization, rehospitalization, or LVAD. (6) Excluded were studies in non-European (except American English) languages and review studies. (7) Although the main results were intended as based on the presence of a multivariate (MV) analysis, we also included articles for critical appraisal that only contained univariate (UV) analysis, to ascertain if any selection bias occurred.

Two reviewers performed title and abstract screening in Covidence [8]. When the decisions of the two reviewers for in- or exclusion were conflicting, the publication was discussed and an agreement was made for further full-text screening, with the possibility of the opinion of a third reviewer. Prior to and during full-text screening, 17 articles were added in Covidence that were known to the authors or were cited in selected articles.

### 3.3. Critical Appraisal

Selected articles were critically appraised with use of the QUIPS-format suited for prognostic studies [9,10]. We edited the QUIPS-format to our interest and identified 8 critical appraisal items for use in our review (see Appendix A). For endpoint-analysis in the HTX-population, we accepted the endpoint of combined mortality and urgent-HTX or LVAD implantation. Since censoring events of HTX—especially when urgent and elective cases were not distinguished—may have influenced the results, we excluded articles that censored events without showing the effects of censoring, either by entering the HTX-candidate as a variable in the prognostic model, or by showing that the delay in performing HTX was not of influence on the outcome.

### 3.4. Data Extraction and Analysis

Studies were categorized into each of the four HF-populations.

Data that were extracted were the following: type of events/endpoints, presence of hemodynamic variables in UV- and/or MV-analysis, and presence of other prognostic predictors (age, sodium, renal function, functional class, ejection fraction, natriuretic peptides, maximal exercise-induced oxygen uptake (VO2-max)). It was noted whether cut-off points or continuous measurements of CI, PCWP and SBP were used. If a variable was measured but not tested in UV- or MV-analysis, the variable was not included for analysis.

### 3.5. Five Different Analyses Were Made

(1)Analysis of UV-variables. The number of studies in which the hemodynamic variable was indicated as a significant predictor, summed up per variable.(2)Main analysis of the number of studies in which the hemodynamic variable was a predictor of prognosis in MV-analysis. A UV-variable, whether non-significant or significant in UV-analysis, could only be included in the main analysis if an MV-analysis was present in the study. A summary analysis was made on quantitative information on prognostic strength of each variable in MV-analysis using hazard ratio (HR), odds ratio (OR) and relative risk (RR).(3)A subanalysis of the hemodynamic predictors in MV-analysis in the four selected HF-populations.(4)Influence of the study quality on the number of studies containing significant or non-significant MV hemodynamic variables of prognosis, by categorizing quality into low- and high-quality categories and by using the Fisher’s exact test. Quality was defined as the number of predefined critical appraisal shortcomings of the study.(5)An additional analysis was performed for comparison between one hemodynamic variable and hemodynamic or other variables in MV-analysis (methods in Appendix A). Three particular predictors of prognosis were added in the comparison between MV-predictors because of their well-known prognostic significance: age, natriuretic peptide levels and VO2-max. Results were evaluated in terms of the number of comparisons in which a hemodynamic variable remained a prognostic predictor in MV-analysis.

## 4. Results

The flow diagram with an inclusion and exclusion process of 529 searched articles is shown in Figure 1. Full-text screening was done of 52 articles using detailed in- and exclusion criteria, excluding 25 articles. All 27 of the remaining articles, containing at least two hemodynamic variables in analysis of prognosis, underwent critical appraisal.

### 4.1. Critical Appraisal

After critical appraisal, seven articles were excluded due to predefined methodological reasons (see Figure 1, under A [11,12,13], B [14,15], C [16] and D [17]). The final 18 articles containing MV-analysis and 2 articles containing only UV-analysis that remained for review are presented in Table 1. For these studies the 8 most crucial critical appraisal-items are summarized in Table 2 while all critical appraisal items can be found in Appendix A.

The patient number was smaller than 200 in 11 out of 20 studies, of which the smallest population comprised 93 HF-patients [20]. In six out of 20 studies, the number of events was low (<50) [19,20,24,25,28,38], and in three out of 20 studies, the ratio of events per number of studied variables was adequate [26,31,36,39]. The outcome measurement was checked externally in only four studies [19,31,34,36].

### 4.2. Included Articles and Data Extraction

For complete data extraction see Appendix A. In all 20 studies, CI and PCWP were measured, while SBP was missing in four studies [24,25,30,34] and not used in UV-analysis in one study [29]. In three studies, CO was used instead of CI [24,30,38]. In the ADHF populations, variables were tested at admission and discharge [36], after 72 h in the hospital [35] or before and after intravenous vasodilator therapy [34].

### 4.3. Univariate Analysis of Hemodynamic Predictors of Prognosis

In nine out of 20 studies, CI or CO was a significant predictor of the outcome in UV analysis [21,22,25,26,27,28,30,33,37]. PCWP was a significant UV-predictor of the outcome in 19 out of 20 studies [19,20,21,22,23,24,25,26,27,28,29,30,31,32,33,34,36,37,38]. Of the 15 studies in which SBP was evaluated in UV analysis, 10 studies demonstrated SBP as a significant predictor [21,22,23,27,28,31,32,34,36,37]. For details on the UV analysis, see Appendix A.

### 4.4. Multivariate Analysis (Main Analysis)

Table 3 summarizes the results of the prognostic value of hemodynamic factors in MV analysis. In none of the 18 studies in which MV-analysis was performed, CI or CO remained significant in MV analysis. Of the nine studies that showed UV significance for CI, seven studies performed an MV-analysis in which CI was not a significant predictor of prognosis [21,22,25,27,30,33,37] and two studies did not further test CI in MV analysis despite UV significance [26,28].

Of the 18 studies that performed an MV-analysis, 17 studies demonstrated PCWP as a UV-predictor [19,20,21,22,23,24,25,27,28,29,30,31,32,33,34,37,38], and one ADHF study showed that PCWP was non-significant in UV analysis [35]. PCWP remained significant in the MV analysis in 10 of 18 studies [19,20,21,22,25,29,31,33,37]; one study did not include PCWP despite UV significance [34]. Quantitative statistical analysis of PCWP was performed in four studies (Table 4).

SBP remained a significant predictor of the outcome in three out of 14 studies that were available for MV analysis [23,31,34]. These 14 studies were composed of nine out of 10 studies with prognostic UV significance for SBP that were further tested in MV-analysis [21,22,23,27,28,31,32,34,37] and five studies that tested SBP as non-significant in UV analysis, but performed an MV analysis on other variables [19,20,33,35,38]. A quantitative statistical analysis of SBP was performed in two studies (Table 4). For details on MV analysis, see Appendix A.

### 4.5. Prognostic Value of Hemodynamic Factors within HF Populations

For results, see Table 3. In MV analysis, CI was not a predictor of prognosis in any of the four populations. PCWP was a significant MV-predictor in four out of four studies in the CHF-population, in four out of 10 studies in the HTX-population, in 0 out of 2 studies in the ADHF-population and 2 of 2 studies in the HFPEF-population. One ADHF study showed that PCWP was non-significant in UV-analysis [35]. In the other ADHF-population, pulmonary artery pressure was assessed as an MV-variable and not PCWP because of their correlation [34]. SBP was not significant in MV-analysis in any of the studies of the CHF- and HFPEF-populations, but was a significant predictor of the outcome in the HTX-population in 2 of 6 studies and in the ADHF-population in 1 of 2 studies. For details on UV-analysis in HF-populations see Appendix A.

### 4.6. Influence of Study Quality Score

Figure 2A shows that CI was not a significant predictor of the outcome in MV-analysis in any study, irrespective of the quality of studies. Figure 2B shows that PCWP was a significant predictor of the outcome in most studies, independently of study quality (Fisher exact test between % of studies with quality 1–4 versus 5–8, *p* = 0.70). In Figure 2C, it is shown that SBP was a predictor of the outcome in some studies, independent of the quality of studies (Fisher exact test between quality 1–4 versus 5–8, *p* = 0.61).

### 4.7. Comparison Between Hemodynamic Factors and Confounders in MV Analysis

Comparisons between the prognostic significance of hemodynamic variables and confounders age, natriuretic peptides and VO2-max are shown in Appendix A. PCWP was the most frequently observed hemodynamic variable, proving to be a significant predictor of prognosis in 15 out of 24 comparisons, versus SBP in 5 out of 21, versus CI in none of the 15 comparisons.

## 5. Discussion

In our systematic review of 20 studies, in which the effects of three hemodynamic factors on prognosis were compared to each other, PCWP emerged as the most frequently observed independent predictor of prognosis. A lower SBP came second as an independent predictor of poor prognosis, but CI was not a predictor of prognosis in any of the studies containing an MV analysis. The results are relevant in four different HF populations and are independent of the quality of the studies. We used the number of studies as a measure of the prognostic strength of the hemodynamic factors, but also retained the individual relative risks of hemodynamic predictors in each study, when available. A meta-analysis was considered as not possible due to a lack of homogeneity in cut-off levels or continuous measures of each hemodynamic variable.

### 5.1. Cardiac Index: No Independent Predictor of Prognosis

In our review, only a single study that did not perform an MV analysis on CI may have argued for CI as a predictor of the outcome [26], against 19 studies that showed the opposite after UV and/or MV analysis. Because UV analysis was always followed by a loss of significance in MV analysis for CI, it is not likely that selection bias for the two studies with only UV analysis has influenced our results.

In two studies that were excluded from our review, resting CI was a significant predictor of prognosis when combined with VO2-max [40,41]. In both studies, all three hemodynamic variables were measured and were significantly related to poor outcome in UV analysis, but only CI was included in MV analysis. In both studies, a low CI can be seen to interact with higher PCWP values for prognosis, and, therefore, PCWP should have been included in MV analysis to be able to answer the question of whether CI would have been an independent predictor of prognosis. Interaction with PCWP was, however, not the only reason for a decrease of the significance of CI in our review, as 11 out of 20 studies already could not demonstrate CI as a univariate predictor of outcome. In addition, three prior studies indicate that short-term changes in CO do not predict the outcome [17,33,36].

### 5.2. Pulmonary Capillary Wedge Pressure: Strongest Predictor of Prognosis

In our review, PCWP was an independent predictor of prognosis in 10 out of 18 studies with MV analyses. Of eight studies that could not demonstrate PCWP as an independent predictor, there were six HTX studies [23,24,27,28,30,32] and two ADHF studies [34,35], of which one study did not further test PCWP despite UV significance because of its interaction with pulmonary artery pressure [34]. What possible interactions with other variables may have caused a loss of significance for PCWP in these studies? In two HTX-studies, PCWP was not significant in MV analysis when VO2-max was present in MV analysis [24,28]. We cannot suggest, however, that VO2-max is always a stronger predictor than PCWP, because in three other studies, PCWP either remained a significant predictor next to VO2-max [31] or was even stronger than VO2-max [22,25]. In another HTX-study, PCWP was found to be non-significant next to a complex variable combining information from CI and PCWP in MV analysis, which is an obvious interaction [30]. Compared to natriuretic peptides, PCWP remained an MV-predictor next to natriuretic peptides in two out of three studies in HTX-populations [25,31], while natriuretic peptides became insignificant next to PCWP in one study [32] (Appendix A).

### 5.3. Systolic Blood Pressure

SBP was the second most important UV-predictor of prognosis in 10 out of 15 studies, and an MV-predictor in three out of 14 studies. What is relevant is that, contrary to the expectation that a higher afterload would be of prognostic disadvantage [8], it was a low SBP and, by inference, a low afterload that was a predictor of mortality without exception. Others have interpreted the finding of low blood pressure influencing prognosis as a result of lower cardiac output influencing prognosis [42], but our review does not support such an interpretation, because SBP was not eliminated from MV analysis when CO was present. Not only is lower blood pressure related to poor prognosis, but in other studies, it is also related to a decreasing SBP independently of baseline SBP [43,44]. The value of a lowering afterload as a treatment goal for all HF patients should, therefore, be questioned, while also recognizing that patients with excessive blood pressure elevations should be treated for hypertension [44].

### 5.4. Population Differences

There were no differences between the HF patient groups as to the predicting ability of CI. We anticipated that CO would be important in the HTX-group because of a more severe reduction in CO, and in UV analysis, this was demonstrated in six out of 11 studies in the HTX-group [25,26,27,28,30,33], but not in any of the HTX-studies after MV analysis. PCWP was an independent predictor of prognosis in all HF populations, and we may argue that it also includes the ADHF population since two of three ADHF studies did not further test PCWP in MV analysis [34,36] and replaced PCWP with pulmonary artery pressure because it was correlated [34].

### 5.5. Hemodynamic Profiles

The hemodynamic profiles of the combined categories of CI and PCWP with specific cut-off values are reported to carry prognostic information [1,14,16,45], although not in all studies [15]. How should we interpret this finding in view of the lack of independent prognostic significance of CI in our review? In two studies [14,45] advocating their use for prognosis, the profiles were made by clinically estimating a decreased CI or an increased PCWP. The only study with specifically measured values of CI and PCWP, did not show survival differences between all profiles [15]. A shortcoming of the profiles in all studies could be the use of cut-off points in PCWP, where information from continuous measurements of PCWP would be as valuable, as shown in Table 4. It is likely that the interaction between CI and PCWP plays a role, such that the lower versus higher CI cut-off value also differentiates patients with lower and higher values of PCWP [34,40,41], while it is the PCWP that predicts prognosis [15,36]. Although a lack of prognostic meaning of CI would apply to all populations we reviewed, an exception must be made for patients in cardiogenic shock, where a low CI and its sequelae are more likely to determine prognosis [16].

### 5.6. Limitations

A limitation is the search strategy itself. We used three search strategies, in which additional MESH-terms or text terms did not further improve the search strategy; we added 10 relevant articles using the references in the searched articles. We only included studies that performed invasive measurements, which could have caused a selection bias. SBP was more often found significant in MV analysis in reviews of non-invasively measured studies [4,5,42]. Another limitation is the use of various combined endpoints in our included studies, but none without mortality.

## 6. Conclusions

A higher PCWP and lower SBP are independent predictors of poor prognosis in HF. In spite of its role in the concept of HF, this review demonstrates that CI is not an independent predictor of prognosis in HF.

## Figures and Tables

**Figure 1 jcm-08-01757-f001:**
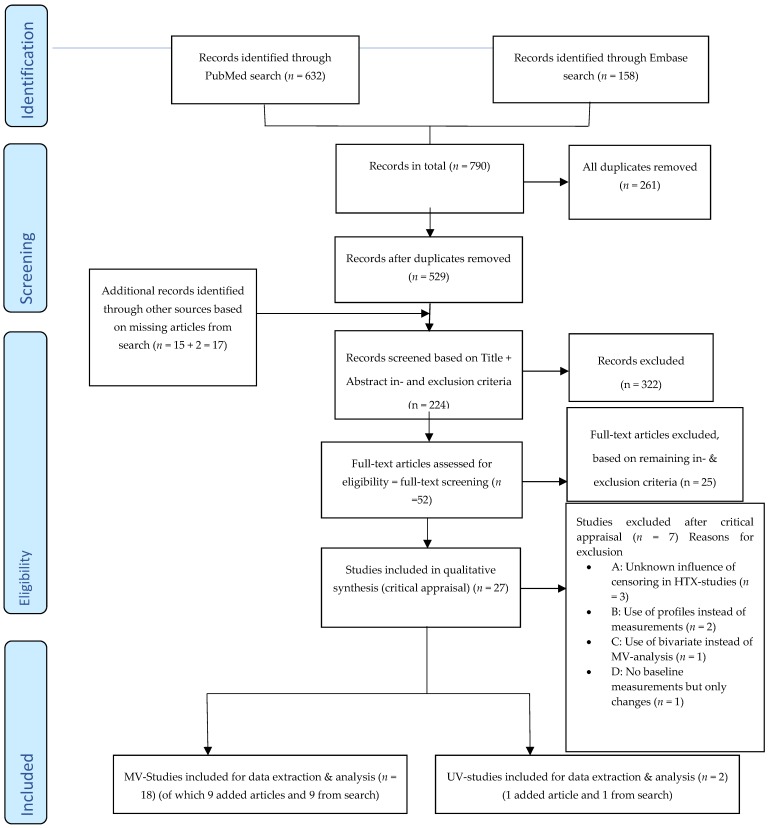
PRISMA flow diagram for selection of studies [18].

**Figure 2 jcm-08-01757-f002:**
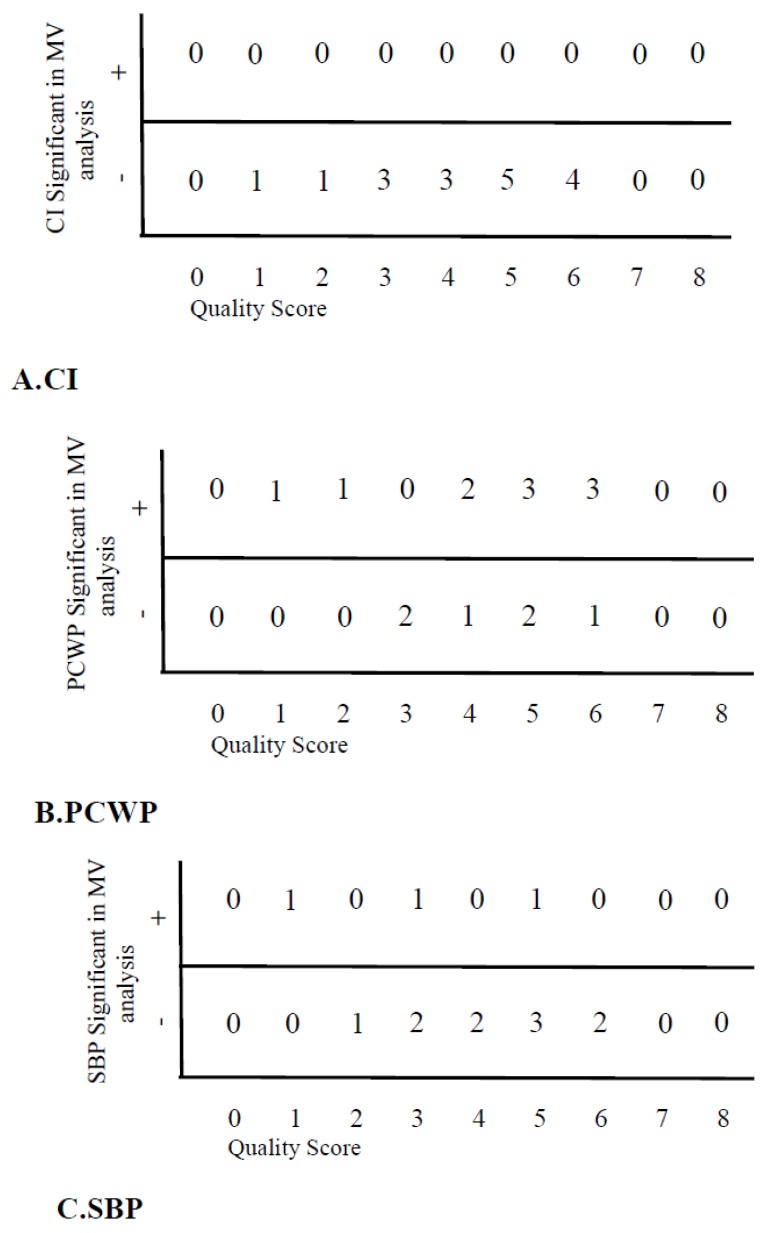
Prognostic significance of hemodynamic variables versus quality of the studies. Number of studies reporting significance for a hemodynamic variable in relation to the quality score of the studies. (**A**) for cardiac output/index (CI), (**B**) for capillary wedge pressure (PCWP), (**C**) for systolic blood pressure (SBP).

**Table 1 jcm-08-01757-t001:** Baseline table of selected studies.

Study Population.	Study Author + Year + Reference	Country	Baseline Year	Study Design	Patient nr	Mean Age (yrs)	Primary Outcome	Follow-Up Duration (months)	Number Events	CI Present (A), Present & Tested in UV (B), or Not Present (X)	PCWP Present (A), Present & Tested UV (B) or Not Present (X)	SBP Present (A), Present & Tested UV (B) or Not Present (X)
CHF	Denardo et al. (2016) [19]	U.S.A.	2008	Prospective Cohort	150	66	mortality, HF hospitalization, rehospitali-zation	12	39 (13 mortality, 26 HF hospitalization/rehospitalization)	B	B	B
	Fauchier et al. (1997) [20]	France	1983	Prospective Cohort	93	51.3 ± 11	Mortality, HTX, cardiomyoplasty	49.5 ± 35.6	23 (14 mortality, 8 HTX, 1 cardiomyoplasty)	B	B	B
	Franciosa et al. (1983) [21]	U.S.A.	1981	Retrospective Cohort study	182	56.5	Mortality	12 ± 10	88	B	B	B
	Guzzetti et al. (2005) [22]	Italy	1991	Prospective Cohort	330	54	progressive pump failure death + urgent HTX	Median 34	108 (62 progressive pump failure death, 17 urgent HTX, 29 sudden death)	B	B	B
HTX	Anguita et al. (1993) [23]	Spain	1986	Prospective Cohort study	130	45 ± 12	mortality, HTX	15 ± 11	93 (46 died, 47 HTX)	B	B	B
	Chomsky et al. (1996) [24]	U.S.A.	1993	Prospective Cohort	185	51 ± 11	mortality and HTX was censored	11 ± 6.9	32 died (and 36 HTX)	B	B	X
	Gardner et al. (2005) [25]	U.K.	2002	Prospective Cohort	97	50.9 ± 10.5	all-cause mortality or urgent HTX	13.2	21	B	B	X
	Ghio et al. (2001) [26]	Italy	1992-1998	Prospective Cohort	377	51 ± 10	Cardiac death or urgent HTX	17.2	140	B	B	X
	Grigioni et al. (2006) [27]	Italy	1996	Retrospective Cohort	196	54 ± 9	CV death and acute HF leading to urgent HTX	24 ± 20	91	B	B	B
	Metra et al. (1999) [28]	Italy	1992	Prospective Cohort	219	55 ± 10	Mortality & urgent HTX	6 (6–54)	38 (32 died and 6 urgent HTX)	B	B	B
	Middlekauff et al. (1991) [29]	U.S.A.	1983	Prospective & Retrospective Cohort study	390	49 ± 12	Total mortality and sudden death (pts undergoing HTX are withdrawn from analysis at the time of surgery)	8.4 ± 10.8	98 (total mortality 98 of which 56 sudden deaths) (HTX = 105)	B	B	A
	Morley et al. (1994) [30]	U.S.A.	1989	Prospective Cohort	138	52 ± 10	Mortality	12	50	B	B	X
	Sachdeva et al. (2010) [31]	U.S.A.	1999	Retrospective Cohort	1215	53 ± 13	Mortality & urgent HTX	24 (31 ± 32)	442 (234 died, 208 urgent HTX)	B	B	B
	Sobieszczańska-Malek et al. (2014) [32]	Poland	2003	Prospective Cohort	559	50.1	Mortality/emergency HTX	21,5	139	B	B	B
	Stevenson et al. (1990) [33]	U.S.A.	1985	Prospective Cohort	152	45 ± 13	overall mortality (including urgent HTX), HTX	12	84 (41 died + 6 urgent HTX, 37 HTX)	B	B	B
ADHF	Aronson et al. (2011) [34]	U.S.A.	1999	Prospective RCT	242	61 ± 14	Mortality	6	61	B	B	B
	Cohn et al. (1984) [35]	U.S.A.	1984	Prospective Cohort	106	54.8	Mortality	1 to 62	60	B	B	B
	Cooper et al. (2016) [36]	U.S.A.	2000	Prospective RCT	151	59	Mortality, cardiovascular hospitalization, HTX	6	103	B	B	B
HFPEF	Dorfs et al. (2014) [37]	Germany	1996	Retrospective Cohort study	355	61.2 ± 11.3	All-cause mortality	112	58	B	B	B
	Goliasch et al. (2015) [38]	Austria	2010	Prospective Cohort study	142	71	Hospitalization for heart failure and/or death for cardiac reason	10	43	B	B	B

**Table 2 jcm-08-01757-t002:** Critical appraisal table, 8 items.

Study Population	Study Author + Year +Reference	Adequate Number of Patients in HF-Population <200 or >200 <200 = −1 & >200 = 0)	Valid and Reliable Measurement of Prognostic Variable; CO Thermodilution or Fick (Thermodilution/Unknown = −1 & Fick = 0)	Outcome Measurement Valid and Reliable: Absence (−1) or Presence (0) of Adjudication Commision/Cheched Externally	Study Attrition (Follow-Up Done in 90% or > of Patient Population) (*N* = −1 & *Y* = 0)	Age (or Other Important Predictive Variable) in MV-Analysis (*N* = −1 & *Y* = 0)	At Least 2 Other UV Significant Variables for HF in UV-Analysis (*N* = −1 & *Y* = 0)	n Events? < 50 or > 50 (<50 = *N* = −1 & >50 = *Y* = 0)	Ratio: n Events/n Variables > 10 (*N* = −1 & *Y* = 0)	Total # of Weak Points on Critical Appraisal
CHF	Denardo et al. (2016) [19]	−1	thermodilution = −1	0	0	−1	−1	−1	−1	−6
	Fauchier et al. (1997) [20]	−1	Unknown = −1	−1	0	−1	0	−1	−1	−6
	Franciosa et al. (1983) [21]	−1	thermodilution = −1	−1	0	0	−1	0	−1	−5
	Guzzetti et al. (2005) [22]	0	unknown = −1	−1	0	−1	0	0	−1	−4
HTX	Anguita et al. (1993) [23]	−1	unknown = −1	−1	0	−1	0	0	−1	−5
	Chomsky et al. (1996) [24]	−1	thermodilution = −1	−1	0	0	0	−1	−1	−5
	Gardner et al. (2005) [25]	−1	thermodilution = −1	−1	0	−1	0	−1	−1	−6
	Ghio et al. (2001) [26]	0	thermodilution = −1	−1	0	−1	0	0	0	−3
	Grigioni et al. (2006) [27]	0	unknown = −1	−1	0	0	0	0	−1	−3
	Metra et al. (1999) [28]	0	thermodilution = −1	−1	0	0	0	−1	−1	−4
	Middlekauff et al. (1991) [29]	0	unknown = −1	−1	0	−1	0	0	−1	−4
	Morley (1994) [30]	−1	thermodilution = −1	−1	0	−1	−1	0	−1	−6
	Sachdeva et al. (2010) [31]	0	unknown = −1	0	0	0	0	0	0	−1
	Sobieszczańska-Malek et al. (2014) [32]	0	thermodilution = −1	−1	0	0	0	0	−1	−3
	Stevenson et al. (1990) [33]	−1	thermodilution = −1	−1	0	−1	0	0	−1	−5
ADHF	Aronson et al. (2011) [34]	0	unknown = −1	0	0	−1	0	0	−1	−3
	Cohn et al. (1984) [35]	−1	thermodilution = −1	−1	0	0	0	0	−1	−4
	Cooper et al. (2016) [36]	−1	thermodilution = −1	−1	0	−1	0	0	0	−4
HFPEF	Dorfs et al. (2014) [37]	0	Fick = 0	−1	0	0	0	0	−1	−2
	Goliasch et al. (2015) [38]	−1	thermodilution = −1	−1	0	0	0	−1	−1	−5

**Table 3 jcm-08-01757-t003:** MV analysis summary table.

Patient Group	CI Number of Studies	PCWP Number of Studies	SBP Number of Studies
CHF	0/4	4/4	0/4
HTX	0/10	4/10	2/6
ADHF	0/2	0/2	1/2
HFPEF	0/2	2/2	0/2
Total of 18 included MV-studies	0/18	10/18	3/14

Number of studies in which a hemodynamic variable was significantly predictive of prognosis. CI = cardiac index, PCWP = pulmonary capillary wedge pressure, SBP = systolic blood pressure, Sign. = significant contributor in MV-analysis, CHF = chronic heart failure, HTX = patients screened for heart transplantation, ADHF = acute decompensated heart failure, HFPEF = heart failure with preserved ejection fraction.

**Table 4 jcm-08-01757-t004:** Multivariate analysis quantitative and qualitative results.

Variable	Variable Measurement Details	Significance	References
CI	Qualitative analysis		
	Continuous	Not Significant	[21,25,27,30,33,37]

	CI ≤ 1.9 L/min/m^2^	Not Significant	[22]
PCWP	Qualitative analysis		
	Continuous, per 1 mmHg	Not Significant	[23,24,28,30]

		Significant	[19,20,21,25,29,33]
	Quantitative analysis		
	Continuous, per 1 mmHg	Sign: HR range = 1.09–1.30	[38,37]
	PCWP ≥ 12 mmHg	Sign: HR= 2.21 (CI (95%) = 1.14–4.17)	[37]

	PCWP ≥ 18 mmHg	Sign: RR= 2.0 (CI (95%) = 1.1–3.5)	[22]

	PCWP ≥ 21 mmHg	Sign: OR= 2.6 (CI (95%) = 1.1–3.0)	[31]
SBP	Qualitative analysis		
	Continuous, increase per 1 mmHg	Not Significant	[21,27,28,32,37]
		Significant	[23]
	SBP ≤ 110 mmHg	Not Significant	[22]
	Quantitative analysis		
	Continuous, per 10 mmHg decrease	Sign: HR = 1.3 (CI (95%) = 1.1–1.5)	[34]
	SBP < 118 mmHg	Sign: OR = 2.8 (CI (95%) = 1.1–7.1)	[31]

CI = cardiac index, PCWP = pulmonary capillary wedge pressure, SBP = systolic blood pressure, HR = hazard ratio, RR = relative risk, OR = odds ratio, CI (95%) = 95% confidence interval.

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
