# Peer review of "Comparison of Hemodynamic Factors Predicting Prognosis in Heart Failure: A Systematic Review"

_jcm, 2019, doi:10.3390/jcm8101757_

Round 1

Reviewer 1 Report

In this review, Aalders et al., reviewed the heart failure literature to find out which amongst cardiac output, preload or afterload are the best prognostic markers in heart failure patients when compared directly. They conclude that cardiac index is not a good independent predictor of prognosis in heart failure. They also conclude that a higher preload and a lower afterload are good independent predictors of poor prognosis in heart failure. 

The study overall is satisfactorily designed. However, the authors should attempt to bring out the novelty and impact of this work in the introduction section of this paper. The question "what does this study add" is also answered in an ambiguous way and needs rewording. Other parts of the paper also require some moderate changes with wording and sentence formation. 

While it is a good practice to repeat the main conclusions of the paper, it becomes challenging to follow when it is repeated multiple times. The authors need to invest a few lines to explain how their proposed strategy to consider decreasing preload and not focus on decreasing afterload or increasing cardiac output impact the clinical practice? How would it affect the present clinical scenario in HF prognosis and how is it better than what has been done until now? Concluding something clinically substantial based on analysis of literature always requires more effort on the authors part to bring home the point in an impactful manner. The authors need to place this work in proper context of the present state-of-the-art scenario for HF prognosis. 

Reviewer 2 Report

 The present systematic review addressed the clinical question which of the three hemodynamic factors predicts prognosis best in patients with heart failure, either cardiac output, preload or afterload (i.e. CI, PCWP and SBP). Methods including definitions and critical appraisal appear to be carefully conformed to a standard procedure in order to assess the question in four different HF-populations. The qualitative study supported the evidence to predict the prognosis of patients with various heart failure, but there are some concerns need to be addressed.

In full-text screening according to the search strategy, a third person would be required for the transparency when the decision of the two reviewers for in- or exclusion were conflicting. Make clear how was an agreement made for the selection of further full-text screening. The results that a higher PCWP and lower SBP are independent predictors of poor prognosis in HF, while not CI seem acceptable. However, there appear to be a gap in the clinical implication regarding a low afterload as a treatment goal for HF. In this study, the optimal criteria of lowering SBP is not principally argued so that the author cannot easily state that treatment goal of heart failure does not consist of decreasing afterload. Make a reference carefully to the clinical practice in patients with HF by discriminating between decreasing excessive levels of SBP as a prognostic predictor and the treatment goal of decreasing afterload.

Author Response

Pleas see the attachment
